# Breaking Boundaries in Retrieval Systems: Unsupervised Domain Adaptation with Denoise-Finetuning

**Che-Wei Chen[1], Ching-Wen Yang[2], Chun-Yi Lin[2], Hung-Yu Kao[2]**
Institute of Medical Informatics[1]
Department of Computer Science and Information Engineering[2]
National Cheng Kung University, Tainan, Taiwan
{Q56104076, P76114511, NE6101050}@gs.ncku.edu.tw
hykao@mail.ncku.edu.tw

## Abstract

Dense retrieval models have exhibited remarkable effectiveness, but they rely on abundant labeled data and face challenges when applied to different domains. Previous domain adaptation methods have employed generative models to generate pseudo queries, creating pseudo datasets to enhance the performance of dense retrieval models. However, these approaches typically use unadapted rerank models, leading to potentially imprecise labels. In this paper, we demonstrate the significance of adapting the rerank model to the target domain prior to utilizing it for label generation. This adaptation process enables us to obtain more accurate labels, thereby improving the overall performance of the dense retrieval model. Additionally, by combining the adapted retrieval model with the adapted rerank model, we achieve significantly better domain adaptation results across three retrieval datasets. We release our code for future research.[1]

## 1 Introduction

The goal of information retrieval (IR) is to enable users to input a query and retrieve relevant passages or documents from the retrieval system. A standard IR system (Matveeva et al., 2006; Liu et al., 2009; Wang et al., 2011; Yang et al., 2019) typically comprises two main stages (refer to Fig. 1):

1. First-stage retrieval model: This model is designed to retrieve a small subset of relevant passages based on the given query.

2. Rerank model: Responsible for reordering the retrieved passages, the rerank model aims to enhance the overall user experience.

Recent advancements in contextualized word embeddings (Liu et al., 2019; Devlin et al., 2019; Lewis et al., 2020) have established dense retrieval

---

[1]https://github.com/eric88525/UDADF

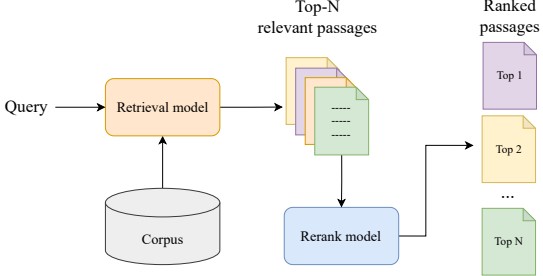

Figure 1: Two-stage information retrieval involves using a retrieval model to select a small subset of passages, which are then reranked using a rerank model.

(Karpukhin et al., 2020; Xiong et al., 2021) as the mainstream approach in information retrieval (IR). This approach effectively addresses the issue of missing words encountered in lexical methods like BM25 (Robertson et al., 1995) or TF-IDF (Christian et al., 2016) and has powerful retrieval ability.

However, the introduction of the BEIR Benchmark (Thakur et al., 2021) has highlighted the limitations of dense models. Surprisingly, traditional lexical-based models like BM25 have exhibited superior performance compared to dense retrieval models in out-of-domain scenarios. Additionally, training dense retrieval models from scratch necessitates a significant amount of domain-specific training data, posing challenges in collecting large-scale data for each specific domain. As a result, there is a growing demand for domain adaptation methods that can enhance model performance without relying on labeled data.

Previous unsupervised domain adaptation approaches, such as Qgen (Ma et al., 2021) and GPL (Wang et al., 2021b), have employed generative models to generate pseudo queries and augment the training data. Qgen and GPL utilized a fine-tuned rerank model (Nogueira and Cho, 2019) on the MSMARCO dataset (Nguyen et al., 2017) to assign relevance scores to the generated queries and passages, creating a pseudo dataset for training the

dense retrieval model. However, they did not adapt the rerank model to the target domain, resulting in potential label imprecision and failure to improve the two-stage retrieval score.

In this paper, we present a novel technique for unsupervised domain adaptation (Fig.2). Our approach focuses on simultaneously adapting the dense retrieval model and rerank model within a two-stage retrieval system. The process begins by generating pseudo queries for target domain passages, treating them as relative pairs. We employ sparse and dense negative miners to identify challenging negatives and create a pseudo dataset by combining queries with relevant and irrelevant passages. The rerank model is then fine-tuned using this dataset.

Since this dataset is not labeled by human, it is likely to contain noisy labels. Therefore, we propose a denoise-finetuning approach that incorporates random batch warm-up and co-regularization learning. Our experimental results validate the effectiveness of this approach in alleviating the impact of noisy labels.

In our domain adaptation approach, we utilize models that have been fine-tuned on the extensive MSMARCO (Nguyen et al., 2017) IR dataset, which serves as a strong foundation for adapting to various domains. Remarkably, our approach outperforms previous domain adaptation methods in both one-stage and two-stage retrieval tasks, showcasing superior performance.

In summary, our pipeline offers the following contributions:

- Full adaptation of both the rerank model and dense retrieval model without requiring target domain labeling data.

- Successful training of the rerank model on the generated dataset, effectively mitigating the influence of noisy labels through random batch warm-up and co-regularization techniques.

- Transfer of domain-adapted knowledge from the rerank model to the dense retrieval model using knowledge distillation (Hofstätter et al., 2020), leading to significant performance improvements in the final two-stage retrieval.

## 2  Related Works

### 2.1  Two-stage Retrieval

Two-stage retrieval (Matveeva et al., 2006; Liu et al., 2009; Wang et al., 2011; Yang et al., 2019)

is a widely adopted approach that combines the strengths of retrieval models and rerank models for effective information retrieval. It has emerged as the preferred pipeline for competitive IR competition tasks (Lassance, 2023; Huang et al., 2023; Zhang et al., 2023).

Rerank models, such as cross-encoder (Nogueira and Cho, 2019) and mono-T5 (Nogueira et al., 2020), demonstrate exceptional performance compared to retrieval models. However, their demanding computational requirements (Khattab and Zaharia, 2020) restrict their practical application to reranking a limited subset of passages.

In this study, we adapt the BERT base retrieval model (Devlin et al., 2019) using the bi-encoder architecture (Reimers and Gurevych, 2019), while the rerank model is implemented using the cross-encoder architecture (Nogueira and Cho, 2019).

The bi-encoder serves as the initial retrieval model, where we input a query $Q$ and a passage $P$ to obtain their respective mean-pooled output vectors $E(Q)$ and $E(P)$. The similarity score between the query and passage is computed by taking the dot product of these embeddings (Eq.1). To optimize efficiency, we pre-compute and store the passage embeddings, allowing for efficient retrieval by encoding the query input and calculating similarity scores.

$$Sim(Q, P) = E(Q) \cdot E(P) \qquad (1)$$

On the other hand, the cross-encoder model treats reranking as a binary classification task. It involves tokenizing each query and passage, denoted as $q^{(1)}, \cdots, q^{(n)}$ and $p^{(1)}, \cdots, p^{(m)}$ respectively. The input sequence then is formed as follows:

$$[[CLS], q^{(1)}, \ldots, q^{(n)}, [SEP], p^{(1)}, \ldots, p^{(m)}] \qquad (2)$$

This sequence is subsequently fed into the BERT model, where the [CLS] embedding is utilized as the input for a single neural network. Cross-encoder exhibits strong performance but has higher latency than the bi-encoder (Hofstätter and Hanbury, 2019). Hence, it is a more suitable choice to pair it with the first-stage retrieval model.

### 2.2  Query Generation

In previous research, docT5query (Nogueira et al., 2019) fine-tuned the T5 (Raffel et al., 2020) model as a query generator using labeled query-passage pairs from the MSMARCO (Nguyen et al., 2017)

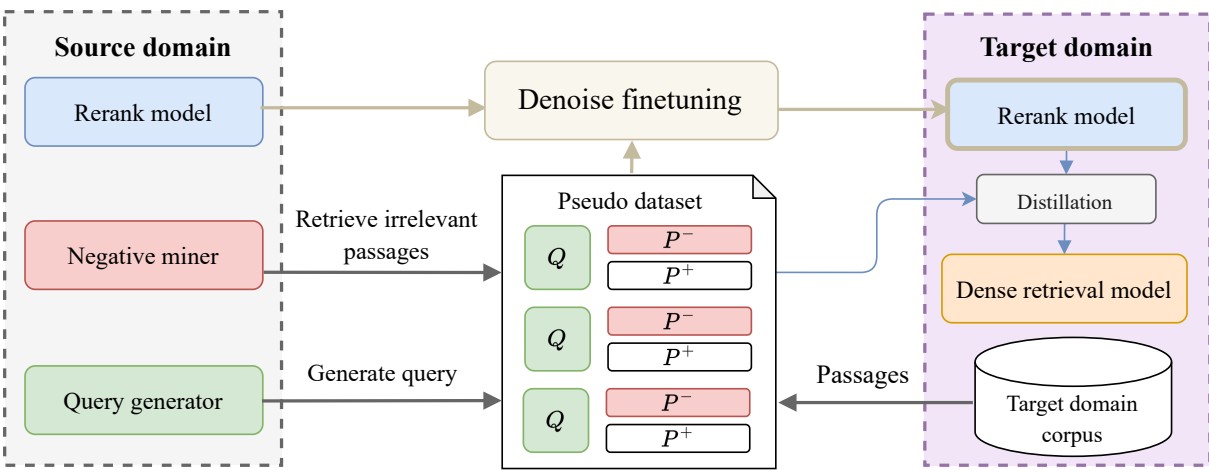

Figure 2: Our adaptation pipeline transfers the rerank model and dense retrieval model from the source domain to the target domain. The process begins with the generation of a pseudo-training dataset, which is then used to fine-tune the rerank model. And then it employs the denoise fine-tuning technique to mitigate the influence of noisy labels. Finally, the knowledge acquired from the rerank model is distilled into the dense retrieval model, completing the adaptation process.

dataset. Their method involved generating questions based on passages and concatenating them with the passages themselves, leading to improved retrieval performance for lexical-based methods.

In the context of domain adaptation, previous methods QGen (Ma et al., 2021) and GPL (Wang et al., 2021b) utilized the generated questions as training data to enhance the dense retrieval model. These methods varied in their labeling approaches, with QGen using binary labels and GPL utilizing a cross-encoder (Nogueira and Cho, 2019) to assign soft labels to query-passage pairs.

Building upon the work of GPL, we further refined the performance of the cross-encoder model in the target domain. By leveraging the adapted cross-encoder, we achieved more precise labeling, resulting in improved performance of the dense retrieval model. Additionally, we employed a two-stage retrieval strategy by combining the adapted cross-encoder with the adapted retrieval model. This innovative approach led to a significant 10% enhancement in NDCG@10 across all three datasets, surpassing the performance of previous methodologies.

## 3 Methodology

Our method, illustrated in Fig.2, achieves domain adaptation for both the retrieval model and rerank model components through three steps: (1) constructing a pseudo training set, (2) denoise-finetuning the rerank model for domain adaptation, and (3) distilling the knowledge from the rerank

model to the dense retrieval model.

### 3.1 Generation of Pseudo Training Dataset

To generate pseudo queries for the passages in a given corpus, we employ a pre-trained T5 model trained on the MSMARCO dataset. To promote diversity, we generate three pseudo queries for each passage, resulting in a total of approximately 300K pseudo queries. For datasets with more than 100K passages, we randomly select 100K passages and generate three pseudo queries for each selected passage. The training data for the target domain comprises around 300K (pseudo query, passage) pairs, denoted as $D_{pseudo} = \{(Q_i, P_i)\}_i$.

### 3.2 Cross-Encoder (Rerank Model) Adaptation

The cross-encoder (Nogueira and Cho, 2019) treats the reranking of passages as a binary classification task. To fine-tune it for the target domain, the training data should consist of (query, passage, label) triplets, where the label is either 1 or 0 to indicate relevance or irrelevance. After obtaining the generated training data $D_{pseudo} = \{(Q_i, P_i)\}_i$, we apply a filter to exclude queries which is shorter than 5 words. This filtering is necessary because we noticed that shorter queries often inquire about the meaning of a word in the format "What is xxx", which does not align with the distribution of the queries in our testing datasets. For a clearer example, please refer to Appendix A.

We leverage a cross-encoder trained on the MS-

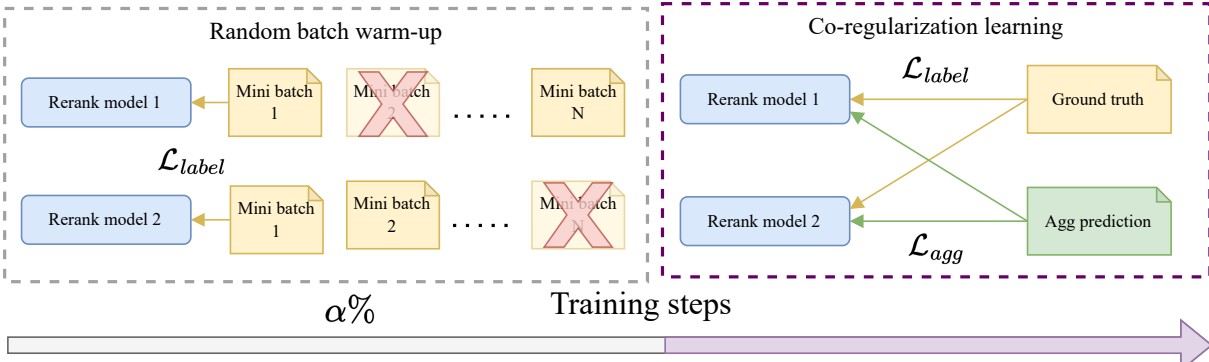

Figure 3: Denoise finetuning: In the random batch warm-up phase, we train two separate cross-encoder models using different data distributions. In the subsequent co-regularization learning stage, these models are jointly optimized with task-specific losses and regulated through an agreement loss. This regularization technique ensures consistent predictions and helps alleviate the influence of noisy labels, effectively preventing overfitting.

MARCO (Nguyen et al., 2017) dataset to assign scores to each (query, passage) pair in $D_{pseudo}$. From these pairs, we extract the top 100k $(Q_i, P_i)$ with the highest cross-encoder prediction scores as positive training samples, forming the positive dataset $D_{pos} = \{(Q_i, P_i, 1)\}_i^{100k}$.

To obtain challenging irrelevant passages, we utilize two retrieval approaches: (1) BM25 and (2) a bi-encoder trained on the MSMARCO dataset. For each query, we retrieve the top-1000 similar passages using both models. From the retrieval pools of these two models, we randomly select one passage from each pool, resulting in two negative samples. This process allows us to create the negative training dataset $D_{neg} = \{(Q_i, P_i, 0)\}_i^{200K}$.

By combining the positive samples from $D_{pos}$ and the negative samples from $D_{neg}$, we construct the final training dataset for the cross-encoder, denoted as $D_{ce} = D_{neg} \cup D_{pos}$. The cross-encoder is trained to predict the probability of relevance between the query and passage pairs, guided by the provided labels in the training data.

This selection process of positive and negative samples ensures that the cross-encoder learns to discriminate between relevant and irrelevant query-passage pairs, enhancing its capability to capture the semantic relationship between queries and passages.

### 3.3 Denoise Finetuning

However, it is probable that the dataset $D_{ce} = \{(Q_i, P_i, y_i)\}_i, y_i \in \{0, 1\}$ contains noisy labels, which can negatively impact the model's performance (refer to Figure.4). These labels can come from (1) low-quality queries generated by the query

generator, (2) false positive passages resulting from imprecise reranking of the cross-encoder, and (3) false negative documents drawn from the top-1000 negative pools.

To address the issue of potentially noisy labels, we propose a learning approach called denoise-finetuning (Fig.3). In our experiments, this approach has been proven to effectively mitigate the impact of incorrect labels.

Taking inspiration from common practices in data science, such as N-fold validation and model ensemble, our objective is for the models to learn from diverse data distributions in the initial stages and subsequently mutually correct each other during the later training phase.

We initialize two cross-encoder models $\{CE_i\}_{i=1}^2$ and employ linear warm-up as the learning rate scheduler. We train our models for $T$ steps, this scheduler gradually increases the learning rate until it reaches its maximum value at $T \times \alpha\%$, where $\alpha$ is a hyperparameter within $[0, 100]$, and subsequently decreases it. During the first $\alpha\%$ of training steps, we randomly discard batches with a probability of $\mathcal{P}$ to allow the two models to learn from different data distributions.

During the later stage of training, we utilize an adapted version of the co-regularization method proposed by Zhou and Chen (2021). This method is inspired by previous studies (Arpit et al., 2017; Toneva et al., 2018) that highlight the delayed learning curves associated with noisy labels. This approach enables both models to learn from the batch data as well as their combined predictions. In this phase, the models benefit not only from the provided labels but also from the aggregated predic-

tions. The influence of co-regularization is controlled by the hyperparameters $\gamma$. The total loss (Eq. 3) consists of two components: $\mathcal{L}_{label}$, which represents the loss based on the provided labels, and $\mathcal{L}agg$, which reflects the loss derived from the joint predictions.

$$\mathcal{L} = \mathcal{L}_{label} + \gamma \cdot \mathcal{L}_{agg} \qquad (3)$$

When a new batch $\mathcal{B}$ is received, the first step is to calculate the task-specific loss $\mathcal{L}_{label}$. In Eq. 4, $\hat{y}_k$ represents the batch prediction of the cross-encoder $CE_k$, and $y$ represents the labels from the batch. We use the binary cross-entropy loss (BCE) for loss calculation.

$$\mathcal{L}_{label} = \frac{1}{2}\sum_{k=1}^{2} BCE(\hat{y}_k, y) \qquad (4)$$

The agreement loss $\mathcal{L}agg$ measures the discrepancy between the model's prediction and the aggregate prediction $agg$. To compute this, we first obtain the aggregate prediction $agg$ by averaging the batch predictions from the two cross-encoder models. Subsequently, we calculate the loss $\mathcal{L}agg$ using the cross-entropy loss.

$$agg = \frac{1}{2}\sum_{i=1}^{2}(CE_k(B)) \qquad (5)$$

$$\mathcal{L}_{agg} = \frac{1}{2}\sum_{k=1}^{2} BCE(\hat{y}_k, agg) \qquad (6)$$

Our experiments demonstrate that by incorporating this co-regularization technique, the models can leverage their shared predictions to enhance their learning and improve overall performance.

### 3.4 Bi-Encoder (Retrieval Model) Adaptation

Knowledge distillation is a widely used method to transfer knowledge from a high-compute model to a smaller and faster model while preserving performance to some extent. Applying this technique to transfer knowledge from a rerank model to a retrieval model is also meaningful.

In a previous study, Hofstätter et al. (2020) proposed a cross-architecture training approach to transfer knowledge from a cross-encoder (Nogueira and Cho, 2019) to various dense retrieval models. They used the cross-encoder as the teacher model to label the margin $\mathcal{M}$ between pairs of (query, relevant passage) and (query, irrelevant passage). By

---

**Algorithm 1:** Denoise Finetuning

**Input:** Dataset $D_{ce} = \{(Q_i, P_i, y_i)\}_i$;
        hyperparameters $T, \gamma, \alpha$;
**Output:** Adapted cross-encoders

Initialize cross-encoders $\{CE_k\}_{k=1}^{2}$
**for** $steps \leftarrow 1$ **to** $T$ **do**
    Select a batch $\mathcal{B}$ from $D_{ce}$
    **if** $step \leq a\% \times T$ **then**
        **for** $k \leftarrow 1$ **to** $2$ **do**
            With probability $1 - \mathcal{P}$:
                $\hat{y}_k \leftarrow CE_k(B)$
                $\mathcal{L}_{label} \leftarrow BCE(\hat{y}_k, y)$
            Update $CE_k$ with $\mathcal{L}_{label}$
    **else**
        Compute predictions:
          $\{\hat{y}_k \leftarrow CE_k(B)\}_{k=1}^{2}$
        Compute label loss: $\mathcal{L}_{label}$ by Eq. 4.
        Compute mean prediction $agg$ by
          Eq.5.
        Compute agreement loss $\mathcal{L}_{agg}$ by
          Eq. 6.
        Total loss: $\mathcal{L} \leftarrow \mathcal{L}_{label} + \gamma \cdot \mathcal{L}_{agg}$.
        Update model parameters with $\mathcal{L}$.

---

utilizing the Margin Mean Squared Error (Margin-MSE) loss and the margin value, they successfully trained the student model (bi-encoder) to discriminate between relevant and irrelevant passages.

The Margin-MSE loss is defined in Eq. 8, where $Q$ denotes the query, $P^+$ denotes the relevant passage, and $P^-$ denotes the irrelevant passage. The term $MSE$ corresponds to the Mean Squared Error loss function (Stein, 1992), while $CE$ and $BE$ represent the output relevant scores of the cross-encoder and bi-encoder, respectively, given a query and passage.

$$\mathcal{M} = CE(Q, P^+) - CE(Q, P^-) \qquad (7)$$

$$\mathcal{L} = MSE(\mathcal{M}, BE(Q, P^+) - BE(Q, P^-)) \qquad (8)$$

Previous experiments conducted by Hofstätter et al. (2020) have demonstrated the effectiveness of ensembling the teacher's scores in improving the accuracy of margin estimation. In our study, we calculate the margin by averaging the scores from both the unadapted cross-encoder and the adapted cross-encoder (referred to as Mix). This approach yields more precise labels for the pseudo queries and passages. For a detailed example, please refer

| Domain | Dataset | #Corpus | #Test query |
|--------|---------|---------|-------------|
| Finance | FiQA-2018 | 57638 | 648 |
| Science | SciFact | 5183 | 300 |
| Bio-Medical | TREC-COVID | 171,332 | 50 |

Table 1: The statistics of the three experimental datasets

to Appendix F.

## 4 Experiment Setup

### 4.1 Datasets

Our experiments are performed on three datasets obtained from the BEIR benchmark (Thakur et al., 2021). Details regarding the test sizes and corpus sizes for each dataset are provided in Table 1 and can be found in Appendix B.

### 4.2 Baselines

In our evaluation, we compare the performance of our adapted cross-encoder and adapted bi-encoder models with previous domain adaptation methods and zero-shot models. Further details about the baselines can be found in Appendix C.

### 4.3 Hyperparameters

We adapt existing models trained on the MS-MARCO (Nguyen et al., 2017) dataset from the source domain to multiple target domains using our domain adaptation method.

**Query Generation:** To generate queries for passages in the target domain, we utilize the T5 query generator [1] with a temperature setting of 1, which has been shown to produce high-quality queries in previous studies (Ma et al., 2021; Wang et al., 2021b).

**Cross-Encoder (Rerank Model) Adaptation:** We adapt the Mini-LM cross-encoder (Wang et al., 2020b)[2] to the target domains. For retrieving hard negatives, we employ BM25 from Elasticsearch and a bi-encoder[3] from Sentence-Transformers. From the top-1000 relevant passages retrieved by each retriever, we select one passage as a hard negative. The input passage is created by concatenating the title and body text. The hyperparameter $\gamma$ in Eq.3 is set to 1. The pseudo training dataset $D_{ce}$

---

[1] https://huggingface.co/BeIR/query-gen-msmarco-t5-large-v1

[2] https://huggingface.co/cross-encoder/ms-marco-MiniLM-L-12-v2

[3] https://huggingface.co/sentence-transformers/msmarco-distilbert-base-v3

consists of 300k samples, with 100k relevant labels and 200k irrelevant labels.

**Denoise-Finetuning:** We allocate the initial $T \times \alpha$ training steps for the random batch warm-up stage, with $\alpha$ set to 0.1. The remaining steps are dedicated to the co-regularization stage. The cross-encoder is trained with a batch size of 32 for 2 epochs. The maximum sequence length is set to 300 for all datasets.

**Bi-Encoder (Retrieval Model) Adaptation:** We use DistilBERT (Sanh et al., 2019) as the pre-trained model for the bi-encoder, following the configuration of GPL (Wang et al., 2021b). For negative retrieval, we employ two dense retrieval models, *msmarco-distilbert*[3] and *msmarco-MiniLM*[4], obtained from Sentence-Transformers. Each retriever retrieves 50 negative passages. From these retrieved passages, we randomly select one negative passage and one positive passage for each training query, forming a single training example. The bi-encoder is trained for 140k steps with a batch size of 32. The maximum sequence length is set to 350 for all datasets.

## 5 Experiment Results and Analyses

### 5.1 Overall Performance

The main results are summarized in Table 2. A comparison of the recall rates for first-stage retrieval is described in Appendix D. We utilize the adapted bi-encoder (BE(w/Ad)) for initial retrieval, retrieving 100 relevant passages per query, and the adapted cross-encoder (CE(w/Ad)) for reranking. This combination yields substantial improvements in NDCG@10 compared to previous domain adaptation methods and zero-shot models. Notably, our adapted bi-encoder outperforms previous query generation-based domain adaptation methods, GPL and QGen, in the first-stage retrieval. We achieve a remarkable 10% performance increase. Note that we do not experiment with BE(Mix) + CE(Mix) setup, since an online system in practice typically comprises one BE and one CE to ensure shorter retrieval time. Introducing Mix as the cross-encoder would double the inference time for reranking, making it impractical for real-world implementation. Therefore, we opted not to explore this particular setup in our experiments.

| Model | FiQA | SciFact | TRECC | Avg. |
|---|---|---|---|---|
| *Zero-shot sparse model* | | | | |
| BM25 | 23.6 | 66.5 | 65.6 | 51.9 |
| docT5query (Nogueira et al., 2019) | 29.1 | 67.5 | 71.3 | 55.9 |
| *Zero-shot dense model* | | | | |
| ANCE (Xiong et al., 2021) | 29.5 | 50.7 | 65.4 | 48.5 |
| ColBERTV2 (Santhanam et al., 2022) | 31.7 | 67.1 | 67.7 | 55.5 |
| *Reranking* | | | | |
| BM25+CE | 35.0 | 68.8 | 75.4 | 59.7 |
| *Pre-training base domain adaptation* | | | | |
| MLM (Gururangan et al., 2020) | 30.2 | 60.0 | 69.5 | 53.2 |
| TSDAE (Wang et al., 2021a) | 29.3 | 62.8 | 76.1 | 56.0 |
| *Query generation base domain adaptation* | | | | |
| QGen (Ma et al., 2021) | 28.7 | 63.8 | 72.4 | 54.9 |
| GPL (Wang et al., 2021b) | 32.8 | 66.4 | 72.6 | 57.2 |
| GPL + CE(wo/Ad) | 36.8 | 68.4 | 76.5 | 60.5 |
| *Ours* | | | | |
| BE(w/Ad) | 33.9 | 69.9 | 74.6 | 59.4 |
| BM25+CE(w/Ad) | 36.7 | 71.1 | 78.6 | 62.1 |
| BE(w/Ad)+CE(w/Ad) | **38.4** | **71.1** | **79.6** | **63.0** |

Table 2: Our approach outperforms previous methods in terms of NDCG@10 for domain adaptation. For reranking by cross-encoder (CE), we utilize the top 100 results from either the bi-encoder (BE) or BM25. For BE(w/Ad), it denotes the bi-encoder (BE) distilled from the Mix cross-encoder (See Table.3).

## 5.2 Impact of Denoise-finetuning

During the evaluation of the denoise-finetuning approach, we conducted tests at intervals of 1000 training steps. In the denoise setting, we reported the model with the higher score. The results depicted in Figure 4 illustrate the effectiveness of denoise-finetuning in stabilizing model training and mitigating the influence of noisy labels, particularly in situations involving diverse data distributions. We compare the two methods using three different random seeds, resulting in six scores for each test point.

## 5.3 Impact of Teacher Models in Distillation

In our distillation experiments, we evaluate three teacher models: (1) adapted cross-encoder, (2) unadapted cross-encoder and (3) Mix model. The Mix model is created by averaging the margin values of models (1) and (2), as described in Section 3.4. We

⁴https://huggingface.co/sentence-transformers/msmarco-MiniLM-L-6-v3

| Teacher | FiQA | SciFact | TRECC | Avg. |
|---|---|---|---|---|
| CE(wo/Ad) | 33.1 | 66.7 | 72.9 | 57.5 |
| CE(w/Ad) | 32.3 | 69.1 | 69.7 | 57.0 |
| Mix | **33.9** | **69.9** | **74.6** | **59.4** |

Table 3: NDCG@10 results obtained using different cross-encoder models as bi-encoder teachers. "Mix" utilizes the average margin score of "CE(wo/Ad)" and "CE(w/Ad)" for training the bi-encoder model.

assess the models every 10k steps and report the best evaluation result.

Interestingly, we observe that while the adapted cross-encoder performs better in reranking, it's distillation result do not surpass the unadapted cross-encoder over all datasets, such as TREC-COVID. After studying the distribution of the margin and some data cases, we found that the unadapted cross-encoder tends to assign a smaller absolute margin score, which means that it exhibits a more conser-

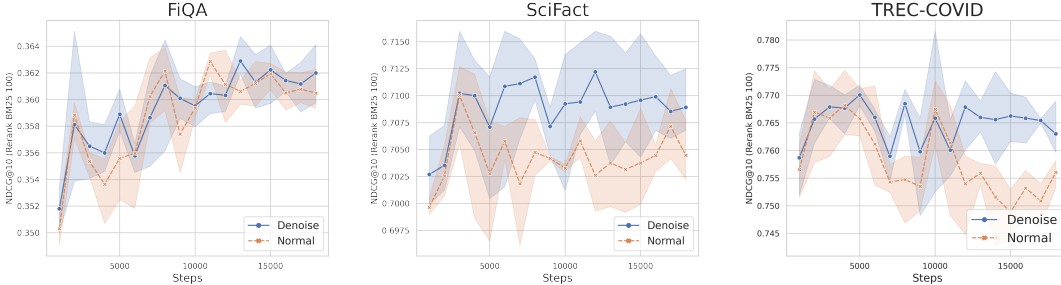

Figure 4: We perform model evaluation on the testing set every 1000 steps during the training of the cross-encoder on the pseudo training dataset. In total, we train the model for 18,750 steps across all datasets, with an allocation of 1870 steps for warm-up and the remaining steps for essential co-regularization. The impact of denoise fine-tuning becomes evident from the second test point, which occurs at approximately 2000 steps.

| Dataset | Method | Reverse% | | |
|---------|--------|------|------|------|
| | | 1 | 5 | 10 |
| FiQA | Normal | 30.3 | 24.7 | 23.2 |
| | Denoise | 33.1 | 26.4 | 24.9 |
| SciFact | Normal | 70.1 | 62.9 | 63.6 |
| | Denoise | 70.1 | 67.5 | 68.4 |

Table 4: The performance of the cross-encoder model was assessed at 18,000 training steps, comparing two training conditions: normal fine-tuning and denoise-finetuning.

vative distinction between relevant and irrelevant passages. We speculate that the adapted model widens the gap between relevant and irrelevant passages, while the unadapted model maintains a certain level of regularization. Consequently, the adapted and unadapted cross-encoders together complement each other and surpass single models. Therefore, we combine both models as Mix and utilize it as the optimal teacher model for knowledge distillation. For a detailed example, please refer to Appendix F.

### 5.4 Extreme Environment

Table 4 presents the results of our experiments on a simulated dataset with significant label noise, demonstrating the effectiveness of denoise finetuning. We inverted $\{1, 5, 10\}\%$ of the labels in the pseudo dataset $D_{ce}$ and report the final test scores. Additionally, we conducted experiments with 20% of the labels inverted, testing with different values of the hyperparameter $\alpha$. Further detailed analysis can be found in Appendix E.

## 6 Conclusion

In conclusion, we have presented a novel approach for unsupervised domain adaptation in information retrieval. Our method addresses the limitations of dense retrieval models and aims to enhance performance in out-of-domain scenarios. By simultaneously adapting the rerank model and dense retrieval model, we achieve significant improvements in two-stage retrieval performance.

Experiment results demonstrate that our domain adaptation approach outperforms previous methods in both one-stage and two-stage retrieval tasks. The fine-tuned rerank model effectively captures domain-specific information, which is then transferred to the dense retrieval model through knowledge distillation.

Overall, our work contributes to the field of unsupervised domain adaptation in information retrieval and provides a valuable framework for improving retrieval performance across diverse domains without the need for domain-specific labeled data.

### Limitations

Our proposed method exhibits superior performance compared to other baselines on FiQA (Finance), SciFact (Science), and TREC-COVID (Bio-Medical) datasets by a significant margin. However, it is important to acknowledge that retrieval datasets from different domains may possess highly diverse distributions of passages and queries. Therefore, further experiments across additional domains are required to comprehensively evaluate the general performance of this method.

## Acknowledgement

This work was supported by the National Science and Technology Council, Taiwan, under Grant NSTC 112-2223-E-006-009. We would like to thank the reviewers for their insightful feedback.

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

| Dataset | Example | |
|---|---|---|
| | Gen | Testset |
| TRECC | what is smus | how does the coronavirus respond to changes in the weather |
| FiQA | what is step2? | what software do you use to track your net worth |
| SciFact | what is mdsc | ATF4 is a general endoplasmic reticulum stress marker. |

Table 5: Example of pseudo queries and testing set queries.

## A   Example of short generated queries

We have noticed that shorter generated queries frequently inquire about the definition or meaning of a word in the format of "What is xxx," which deviates from the distribution found in our testing dataset. See Table 5 for the comparison of a generated query and a query from the corresponding testing dataset.

## B   Dataset

We evaluate the performance of our method on three datasets from the BEIR benchmark. To ensure a fair evaluation, we utilize the datasets and evaluation code [5] provided by Thakur et al., 2021.

The **FiQA** dataset (Maia et al., 2018) focuses on the financial domain and utilizes StackExchange posts from 2009 to 2017 as its corpus.

The **TREC-COVID** dataset(Voorhees et al., 2021) is based on the CORD-19 (Wang et al., 2020a) challenge and includes recent publications and historical research on COVID-19 and related coronaviruses.

The **SciFact** dataset (Wadden et al., 2020) comprises evidence-containing abstracts that correspond to expert-written scientific claims. Each claim is labeled, and the dataset also includes rationales that provide supporting evidence for the labels.

We selected these three datasets for the following reasons:

**Accessibility**: All three datasets are publicly available and open for use.

**Objective Evaluation**: We employed the BEIR toolkit, utilizing its provided corpus, queries, qrels, and validation methods, ensuring a fair and unbiased evaluation.

**Diverse Domains**: Our chosen datasets span diverse domains, including finance, scientific abstracts, and COVID-19 research, representing areas of significant relevance.

## C   Baseline

We compare our approach with previous zero-shot and domain adaptation approaches. For GPL, MLM, Qgen and TSDAE, we utilize the data provided from Wang et al. (2021b). For docT5query, we use the data provided by the Thakur et al. (2021). For BM25, we employ the Anserini toolkit [6] with a pre-built index.

**Zero-shot sparse model**: The BM25 (Robertson et al., 1995) and docT5query (Nogueira et al., 2019) models represent the sparse approach. The docT5query model utilizes a T5 model (Raffel et al., 2020) to generate queries and append them to the passages, thereby improving the retrieval performance.

**Zero-shot dense model**: ANCE (Xiong et al., 2021) is a bi-encoder that utilizes an Approximate Nearest Neighbor (ANN) index of the corpus to generate challenging negative examples. During model fine-tuning, the index is updated in parallel to select these instances. On the other hand, ColBERTV2 (Santhanam et al., 2022) computes contextualized embeddings at the token level, which offers better robustness compared to whole-sentence-based approaches.

**Pre-training**: MLM (Gururangan et al., 2020) and TSDAE (Wang et al., 2021a) enable unsupervised training of the model in the target domain, facilitating subsequent fine-tuning for downstream tasks.

**Question generation domain adaptation**: QGen (Ma et al., 2021) and GPL (Wang et al., 2021b) leverage generated questions as training data to enhance the dense retrieval model. These methods differ in their labeling approaches, with QGen using binary labels, and GPL utilizing a cross-encoder (Nogueira and Cho, 2019) to assign soft labels to query-passage pairs.

## D   Recall Performance

In the context of two-stage retrieval, the recall rate of the first retrieval model holds paramount significance. This is because achieving maximum recall of relevant articles is essential to enable the subsequent rerank model for effective sorting. In Table

---

[5] https://github.com/beir-cellar/beir.git

[6] https://github.com/castorini/anserini.git

| Model/Dataset | FiQA | SciFact | TRECC |
|---|---|---|---|
| BM25 | 53.9 | 92.5 | 47.1[*] |
| GPL | **63.7** | 89.5 | 53.8[*] |
| QGen | 61.8 | 89.3 | 45.6[*] |
| BE(w/ad) | 63.1 | **91.6** | **54.7**[*] |

Table 6: Comparison between Recall@100 of our approach with BM25 and a previous domain adaptation method. The '*' denotes capped recall@k.

| Dataset / $\gamma$ | 0 | 1 | 2 | 5 | 10 | 20 |
|---|---|---|---|---|---|---|
| FiQA | 25.8 | 29.0 | 27.7 | 30.1 | 33.4 | 35.0 |
| SciFact | 65.1 | 62.8 | 66.4 | 67.4 | 68.9 | 71.4 |

Table 7: Reranking BM25 top 100 results with ranking model in the latest checkpoint evaluation on test set.

6, we conduct a comparison between the adapted bi-encoder and other domain adaptation methods in terms of Recall@100.

Due to the fact that the TREC-COVID dataset contains over 100 instances with relevance annotations for each query, utilizing the standard version of recall would result in very low values, as the number of relevant passages always exceeds the parameter 'k'. Hence, we adopt the capped recall@k (Eq.10) as a replacement measure, the same as BEIR Benchmark (Thakur et al., 2021).

$$\text{Recall@k} = \frac{1}{N} \sum_{i=1}^{N} \frac{\text{TP}_i}{\text{GT}_i} \quad (9)$$

$$\text{Capped Recall@k} = \frac{1}{N} \sum_{i=1}^{N} \frac{\text{TP}_i}{\min(k, \text{GT}_i)} \quad (10)$$

where:

$N$ : Total number of queries.

$\text{TP}_i$ : Number of true positives for retrieval instance i.

$\text{GT}_i$ : Number of ground truth positive items for retrieval instance $i$.

$k$ : Cutoff value for recall calculation.

## E  Extremely Noisy Setting

In our experiments, we intentionally create noisy labels by inverting 20% of the labels in the pseudo dataset $D_{ce}$. This allowed us to simulate an extremely noisy dataset and evaluate the performance

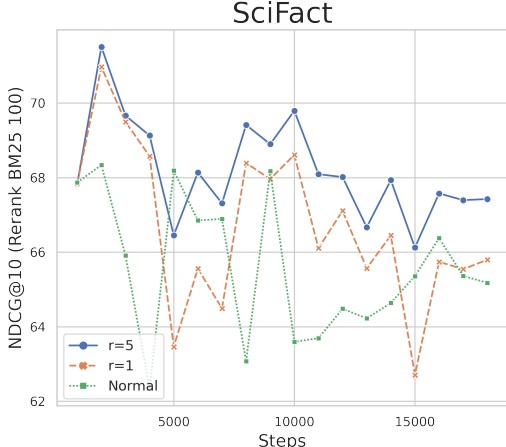

Figure 5: Experimental results with reversed 20% labels on the SciFact dataset.

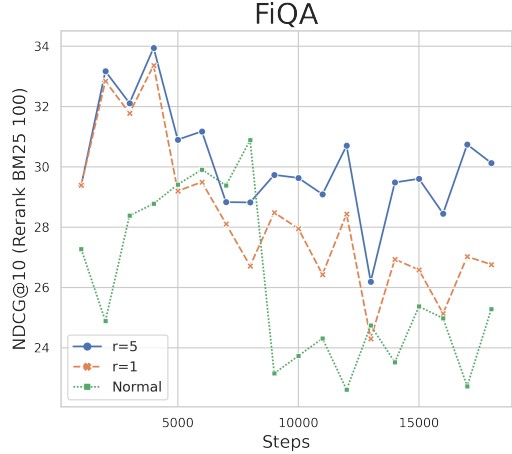

Figure 6: Experimental results with reversed 20% labels on the FiQA dataset.

of the denoise-finetuning method. We conducted tests every 1000 training steps to track the model's fine-tuning progress. The results presented in Figure 5 and Figure 6 demonstrate the impact of the hyperparameter $\gamma$ (defined in Eq. 3). Table 7 shows the last checkpoint rerank performance. These findings highlight the importance of carefully tuning this parameter to achieve optimal performance and minimize the influence of label noise.

## F  Margin from different teacher models

In Table 8 and Figure 7, different models exhibit variations in margin predictions for positive and negative samples across different datasets. Hence, we can leverage these differences to complement each other's shortcomings.

| **CE(wo/Ad)**: -0.41 **CE(w/Ad):** -11.56 **Mix:** -5.98 | **CE(wo/Ad)**: 4.63 **CE(w/Ad):** -9.15 **Mix:** -2.26 |
|---|---|
| **Q**: how to calculate the rent you can get from owning your house

**P+** ...Yes, you should certainly compare the monthly rent to what your mortgage payments would be, as you have done. Yes, you should consider how long you might live there. If you do move out, how difficult will it be to sell the house, given the market conditions in your area? If you try to rent it, how difficult is it to find a tenant, and what rent could you expect to receive? Speaking of moving out and renting the place: Who will manage the property and do maintenance? ...

**P-** In order to arrive at a decision you need the numbers: I suggest a spreadsheet. List the monthly and annual costs (see other responses). Then determine what the market rate for rental. Once you have the numbers it will be clear from a numbers standpoint. One has to consider the hassle of owning property from a distance, which is not factored into the spreadsheet | **Q**: why is honesty important in business

**P+** ...I refuse to be dishonest in business and am lucky to work for senior management that shares that view. We win some, we lose some. We are as transparent as possible, keep our commitments, and seek deals that drive value for all parties. Sometimes I am deeply frustrated by the lack of integrity of others and the benefits they seem to reap. But my reputation is excellent, my clients know they can take me at my word...

**P-** Even if there's nobody outside who needs to see it right away (investors, etc.), an honestly-written business plan is a valuable exercise. If done well (and preferably with external guidance) it forces you to think all the way through your idea and make sure your bases are covered. |

Table 8: In the example on the left, CE(w/Ad) demonstrates better performance, whereas in the example on the right, CE(wo/Ad) outperforms. Various models showcase diverse margin predictions for positive and negative samples across distinct datasets. Thus, we can exploit these variances to compensate for each model's limitations.

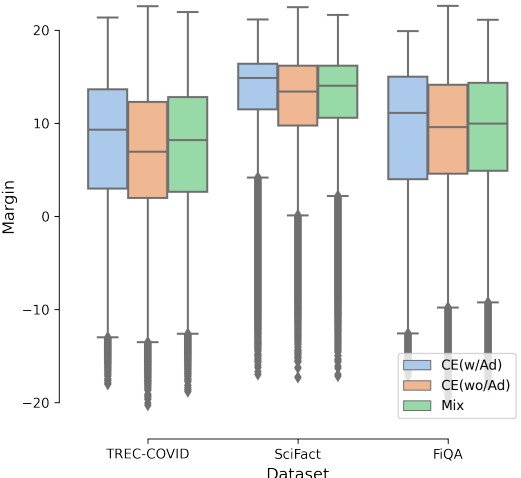

Figure 7: The margin values of using adapted cross-encoder, unadapted cross-encoder, and mixed margin.