# OpenReview forum: "Breaking Boundaries in Retrieval Systems: Unsupervised Domain Adaptation with Denoise-Finetuning"
_EMNLP/2023/Conference — EMNLP 2023 Findings_

### Official Review · Reviewer_XVNc · 2023-08-04

**Soundness:** 2

**Excitement:**

3: Ambivalent: It has merits (e.g., it reports state-of-the-art results, the idea is nice), but there are key weaknesses (e.g., it describes incremental work), and it can significantly benefit from another round of revision. However, I won't object to accepting it if my co-reviewers champion it.

**Paper Topic And Main Contributions:**

The authors proposed a rerank method to improve unsupervised domain adaptation in dense retrieval tasks.
1. The authors leverage target domain data into rerank and retrieval modeling.
2. A denoise algorithm is introduced to increase the dense retrieval performance.
3. The authors enhance the dense retrieval model with knowledge distillation from rerank model.



**Questions For The Authors:**

Question A:
Noisy Label Handling: The paper does not delve into how they handle cases with high label noise (>10%). Although the denoise fine-tuning method is a good step, a more explicit technique, such as a noise-robust loss function, might be beneficial to tackle extremely noisy labels.
Question B:
I found the authors pre-trained their model on the MS-MARCO dataset for adaptation to multiple target domains. However, using a single source for domain adaptation might limit the model's ability to generalize to other unseen domains.

**Reasons To Accept:**

1. The paper presents an approach to unsupervised domain adaptation in information retrieval.
2. The experimental results present that the proposed method significantly outperforms existing methods on various datasets (FiQA, SciFact, and TREC-COVID), indicating the effectiveness of the proposed method.

**Reasons To Reject:**

1. The compared baselines are not strong. Most of them are before 2022.
2. The proposed method involves multiple steps: generating queries, adapting cross-encoder and bi-encoder, and denoise fine-tuning. All modules are with different hyperparameters. This complexity can lead it difficult to reproduce and can also increase computational costs.

**Reproducibility:**

2: Would be hard pressed to reproduce the results. The contribution depends on data that are simply not available outside the author's institution or consortium; not enough details are provided.

**Reviewer Confidence:**

2: Willing to defend my evaluation, but it is fairly likely that I missed some details, didn't understand some central points, or can't be sure about the novelty of the work.

**Typos Grammar Style And Presentation Improvements:**

Suggestions for authors:
1. Evaluation Metrics: The paper mainly focuses on NDCG@10 for evaluation, which may not fully reflect the model's performance, more evaluation metrics should be introduced.
2. Experimentation with Transformer-based Models: The authors could investigate other cutting-edge transformer models for the query generation task.
3. Teacher Model Selection: While the paper successfully demonstrated the usage of a Mix model as a teacher for distillation, it relies on empirical observations. More theoretical exploration into why the Mix model performs better could make the method more robust and interpretable.

---

> ### Author Rebuttal · Authors · 2023-08-28
>
> Thank you for your valuable suggestions and comments, which have been addressed below.
>
> > The compared baselines are not strong. Most of them are before 2022
>
>
> Our primary focus in this study is domain adaptation, specifically comparing against the state-of-the-art methods in the domain adaptation methods, such as QGen[RF1] and GPL[RF2]. While there are indeed more complex rerank and retrieval models available, these often involve intricate training techniques, extra datasets, and model architectures.
>
> Our intention with this work was to demonstrate the effectiveness of our approach using even the simplest models, namely cross-encoder and bi-encoder, for domain adaptation. By doing so, we aim to showcase that our method remains effective even with basic models.
>
> [RF1] Yang, Ji Ma Ivan Korotkov Yinfei, and Keith Hall Ryan McDonald. "Zero-shot Neural Passage Retrieval via Domain-targeted Synthetic Question Generation."
> [RF2] Wang, Kexin, et al. "GPL: Generative Pseudo Labeling for Unsupervised Domain Adaptation of Dense Retrieval." Proceedings of the 2022 Conference of the North American Chapter of the Association for Computational Linguistics: Human Language Technologies. 2022.
>
> > The proposed method involves multiple steps: generating queries, adapting cross-encoder and bi-encoder, and denoise fine-tuning. All modules are with different hyperparameters. This complexity can lead it difficult to reproduce and can also increase computational costs.
>
>
> **Computational costs**
>
> You've correctly identified that our method involves multiple steps and numerous hyperparameters. We acknowledge that finding suitable hyperparameters can be challenging due to the complexity. While it's true that our approach involves more steps than GPL, we believe these additional costs are justified for the following reasons:
> 1. We cannot guarantee that an unadapted cross-encoder will accurately annotate the target domain.
> 2. The integration of cross-encoders from different domains during denoise fine-tuning adds diverse perspectives, which could be more beneficial than traditional co-regularization methods. For instance, in scenarios where we're building a retrieval system for a new disease, denoise fine-tuning with related diseases' retrieval systems could yield mutual benefits. The aggregated predictions (Eq.7) from multiple integrated models might enhance performance.
> 3. The majority of the added cost is incurred during the training process, with minimal computational cost increase during actual inference.
>
> **About reproduce**
> We will release our code along with the generated queries and pseudo training dataset.
>
>
> > Question A: Noisy Label Handling: The paper does not delve into how they handle cases with high label noise (>10%). Although the denoise fine-tuning method is a good step, a more explicit technique, such as a noise-robust loss function, might be beneficial to tackle extremely noisy labels
>
> In response to your question regarding handling high label noise, we have conducted additional experiments to investigate the performance of denoise fine-tuning under extreme noise conditions. We created a pseudo dataset with 20% flipped labels to simulate extremely noisy labels. We adjusted the impact of denoise fine-tuning using $\gamma$ in Eq.5 to observe its effects on the final checkpoint's performance on the test set's BM25 top 100 results' NDCG@10 scores.
>
> The results indicate that denoise fine-tuning remains effective even in extremely noisy conditions. This demonstrates the robustness of our approach in handling label noise.
>
>
> | Dataset / $\gamma$ | 0 | 1 |  2 |  5 | 10 | 20 |
> | -------- | -------- | -------- | - | -------- | -------- | -------- |
> | FiQA     | 25.8     | 29.0     |27.7 | 30.1| 33.4| **35.0** |
> | SciFact     | 65.1     | 62.8     | 66.4| 67.4|68.9 | **71.4**|
>
>
> > Question B: I found the authors pre-trained their model on the MS-MARCO dataset for adaptation to multiple target domains. However, using a single source for domain adaptation might limit the model’s ability to generalize to other unseen domains.
>
> Our model's source domain for domain adaptation is MSMARCO[RF3] , which is currently one of the largest information retrieval datasets. This choice allows us to leverage various supervised models for domain adaptation. Other works in the field have also begun from MSMARCO as a starting point for domain adaptation, justifying our approach.
>
> What distinguishes our approach from previous domain-adaptation methods, such as GPL, is the potential for integrating cross-encoders from different source domains during denoise fine-tuning and utilizing them for distillation in the bi-encoder.
>
> [RF3] Nguyen, Tri, et al. "Ms marco: A human-generated machine reading comprehension dataset." (2016).
>
> > Evaluation Metrics: The paper mainly focuses on NDCG@10 for evaluation, which may not fully reflect the model’s performance, more evaluation metrics should be introduced.
>
> We will include additional metrics recall@100 in our evaluation process. In a two-stage retrieval system, the recall rate of the first stage is  crucial, as successful retrieval in the first stage is a prerequisite for subsequent ranking by rerank models.
>
> We have compared the first stage retrieval model's recall@100, and our approach exhibits higher recall@100 in SciFact and TREC-COVID datasets compared to QGen and GPL, which signifies the effectiveness of our approach in the initial retrieval stage.
>
>
> | Method / Dataset | FiQA | SciFact | TRECC|
> | -------- | -------- | -------- | -|
> |   QGen   | 61.8     | 89.3     | 45.6|
> |   GPL   | **63.7**     | 89.5     | 53.8|
> | BE(w/Ad)     | 63.1     | **91.6**    | **54.7** |
>
>
>
> > Experimentation with Transformer-based Models: The authors could investigate other cutting-edge transformer models for the query generation task.
>
> In the future, we are interested in exploring the use of ChatGPT as a query generator. However, due to budget constraints and the significant time investment required for generation, we temporarily adopted the same query generator as our previous work. This suggestion is intriguing, and I agree that using different query generator models could provide distinct insights into the target domain.
>
> > Teacher Model Selection: While the paper successfully demonstrated the usage of a Mix model as a teacher for distillation, it relies on empirical observations. More theoretical exploration into why the Mix model performs better could make the method more robust and interpretable.
>
> While the large number of label relations, consisting of 4,480,000 (140,000*32) (Query, positive passage, negative passage) triples used for cross-encoder knowledge distillation to the bi-encoder, lacks precise annotated data for statistical analysis, we can hypothesize several factors contributing to the success of the Mix model:
> + The cross-encoder adapted through domain adaptation demonstrates enhanced ranking capability, highlighting its proficiency in distinguishing relevant and irrelevant passages concerning the query.
> + Our case study reveals that both models excel in different sample scenarios.
>
> We will enhance the interpretability of our method by incorporating case studies specific to each of the three datasets in future versions of the paper. This will provide a more comprehensive explanation of the rationale behind the Mix model's effectiveness.
>
>
> **To Reviewer:**
>
> Once again, we appreciate your constructive feedback and suggestions, which helps us to enhance the comprehensiveness of our research.

---

### Official Review · Reviewer_kWur · 2023-08-04

**Soundness:** 3

**Excitement:**

2: Mediocre: This paper makes marginal contributions (vs non-contemporaneous work), so I would rather not see it in the conference.

**Paper Topic And Main Contributions:**

This paper proposed a novel approach for unsupervised domain adaptation in information retrieval. The main contribution of this paper is the proposal of a method that addresses the limitations of dense retrieval models and aims to enhance performance in out-of-domain scenarios. By simultaneously adapting the rerank model and dense retrieval model, the authors achieve significant improvements in two-stage retrieval performance. The paper also provides a comprehensive evaluation of the proposed method on three different datasets from different domains, demonstrating its superior performance compared to other baselines. Overall, this paper contributes to the field of unsupervised domain adaptation in information retrieval and provides a valuable framework for improving retrieval performance across diverse domains without the need for domain-specific labeled data.

**Reasons To Accept:**

1. The novel approach to unsupervised domain adaptation with denoise-finetuning, which significantly improves the performance of dense retrieval models.
2. A comprehensive evaluation of the proposed method on three different datasets from different domains, demonstrating its superior performance compared to other baselines.

**Reasons To Reject:**

1. The description in the related work (Section 2) is somewhat lengthy, which covers more than one page.
2. The reason why choose the three datasets is not clearly clarified.

**Reproducibility:**

2: Would be hard pressed to reproduce the results. The contribution depends on data that are simply not available outside the author's institution or consortium; not enough details are provided.

**Reviewer Confidence:**

1: Not my area, or paper was hard for me to understand. My evaluation is just an educated guess.

---

> ### Author Rebuttal · Authors · 2023-08-28
>
> Thank you for your feedback, which we greatly appreciate.
>
> > The description in the related work (Section 2) is somewhat lengthy, which covers more than one page.
>
> We understand your concern regarding the length of the related work section (Section 2). During the initial drafting phase, we aimed to make our work accessible to scholars outside the Information Retrieval (IR) domain. To achieve this, we aimed for a comprehensive description to provide context to non-IR researchers. We will carefully consider your input and decide whether to trim the content while ensuring the essential context is retained.
>
> > The reason why choose the three datasets is not clearly clarified.
>
>
> Regarding the choice of the three datasets, we have several reasons for our selection:
> + **Accessibility:** All three datasets are publicly available and open for use.
> + **Objective Evaluation:** We employed the BEIR toolkit, utilizing its provided corpus, queries, qrels, and validation methods, ensuring a fair and unbiased evaluation.
> + **Diverse Domains:** Our chosen datasets span diverse domains, including finance, scientific abstracts, and COVID-19 research, representing areas of significant relevance.
>
> We will incorporate these explanations into the paper to clarify our rationale for dataset selection.
>
> Once again, we sincerely thank you for your detailed review and hope that our responses encourage you to reconsider the evaluation of our work.

---

### Official Review · Reviewer_X5Zx · 2023-08-17

**Soundness:** 3

**Excitement:**

3: Ambivalent: It has merits (e.g., it reports state-of-the-art results, the idea is nice), but there are key weaknesses (e.g., it describes incremental work), and it can significantly benefit from another round of revision. However, I won't object to accepting it if my co-reviewers champion it.

**Paper Topic And Main Contributions:**

This paper investigates domain adaptation of the two-stage retrieval system. The authors first construct a pseudo training set using an initial generation model and two negative sample retrieval strategies. Then, they propose a denoise fine-tuning method to train the rerank models, where a key co-regularization technique is introduced to constrain the loss of different sub-rerank models. Finally, they use the rerank model as a teacher and update the retrieval model using knowledge distillation. The authors carefully experimentally verify the effectiveness of their method in domain adaptation.

**Questions For The Authors:**

1. How does the proposed denoise fine-tuning method compare to R-Drop?

2. Is the validation set available? If not, is the performance reported for the last checkpoint?

**Reasons To Accept:**

1. The paper is easy to understand, and the entire method appears to be reliable.

2. The experiments show a significant improvement compared to previous methods, e,g., QGen, GPL.

**Reasons To Reject:**

1. The relationship between the proposed co-regularization and denoising is not yet clear. It seems to only constrain the losses of the two sub-models, reducing overfitting and making training more stable.

2. The experimental section lacks discussion on some necessary details, such as: how to choose the two key parameters alpha and P? which rerank model should be chosen for knowledge distillation? would it be better to fine-tune with more rerank models?

3. No repeated experiments were conducted to eliminate the influence of randomness.

**Reproducibility:**

4: Could mostly reproduce the results, but there may be some variation because of sample variance or minor variations in their interpretation of the protocol or method.

**Reviewer Confidence:**

4: Quite sure. I tried to check the important points carefully. It's unlikely, though conceivable, that I missed something that should affect my ratings.

**Typos Grammar Style And Presentation Improvements:**

Section 2.3 is related to knowledge distillation, and it would be better to merge it into Section 3.4.

---

> ### Author Rebuttal · Authors · 2023-08-28
>
> Thank you for your valuable suggestions and comments, which have been addressed below.
>
> > The relationship between the proposed co-regularization and denoising is not yet clear. It seems to only constrain the losses of the two sub-models, reducing overfitting and making training more stable.
>
> We introduce multiple models to collaboratively predict a shared label, allowing all models to learn together. We believe that this shared label becomes closer to the true underlying label because:
> + Noisy labels exhibit delayed learning curves [RF1], indicating that models might not have learned these errors at this stage.
> + The concept of predicting shared labels draws inspiration from prior research [RF2], and using multiple models for co-regularization has been explored in related studies to mitigate the impact of mislabeling.
>
> [RF1] Arpit, Devansh, et al. "A closer look at memorization in deep networks." International conference on machine learning. PMLR, 2017.
>
> [RF2] Zhou, Wenxuan, and Muhao Chen. "Learning from Noisy Labels for Entity-Centric Information Extraction." Proceedings of the 2021 Conference on Empirical Methods in Natural Language Processing. 2021.
>
>
> What distinguishes our approach from previous co-regularization  methods, is the potential for **integrating** cross-encoders from different source domains.
> + For instance, in scenarios where we're building a retrieval system for a new disease, denoise fine-tuning with related diseases' retrieval systems could yield mutual benefits. The aggregated predictions (Eq.7) from multiple integrated models might enhance performance.
>
> > The experimental section lacks discussion on some necessary details, such as: how to choose the two key parameters alpha and P? which rerank model should be chosen for knowledge distillation? would it be better to fine-tune with more rerank models?
>
>
> The choice of the two parameters alpha and P is challenging due to their sensitivity to various factors like training data size and batch size. However, we do believe it's meaningful to explore the coefficient $\gamma$ in Eq.5, as it directly influences denoise fine-tuning.
>
> We do a experimented with a variety of  $\gamma$ to investigate the impact of denoise fine-tuning. In a simulated extreme noisy environment, where 20% of labels were flipped, we observed that denoise fine-tuning consistently led to improvements in the final checkpoint's NDCG@10 scores, in comparison to models trained without denoise-finetuning.
>
> | Dataset / $\gamma$ | 0 | 1 |  2 |  5 | 10 | 20 |
> | -------- | -------- | -------- | - | -------- | -------- | -------- |
> | FiQA     | 25.8     | 29.0     |27.7 | 30.1| 33.4| 35.0 |
> | SciFact     | 65.1     | 62.8     | 66.4| 67.4|68.9 | 71.4|
>
>
> To address your concern about the rerank model for knowledge distillation, we chose the one that performed the best on the test set. Using more rerank models for fine-tuning has been shown to not work well in previous research [RF1], and it also results in increased training costs.
>
> > No repeated experiments were conducted to eliminate the influence of randomness.
>
> We acknowledge this concern and have taken steps to address it. We conducted experiments with three different random seeds in our laboratory setting, performing evaluations every 1000 training steps. In the final version of the paper, we will provide a comprehensive representation of the results by including the evaluation scores from all three seeds at the same step intervals. This will allow readers to observe the outcomes of denoise fine-tuning and normal fine-tuning for a total of six scores (three for denoise fine-tuning and three for normal fine-tuning).
>
>
> > How does the proposed denoise fine-tuning method compare to R-Drop?
>
> This is a great question! We are also curious about this. To compare out work with R-Drop[RF1]:
> + We followed the implementation detailed in the R-Drop paper but changed the KL divergence to binary cross-entropy loss due to cross-encoder's binary output.
> + We conducting evaluations on the test set every 1000 training steps during finetuning, using BM25 top 100 passages to rerank.
> + Our reported results highlight the best NDCG@10 scores achieved during the training process.
> + In accordance with Section 4.4 of the R-Drop paper, we set the dropout rate to 0.3 and explored the α parameter within the search range {3, 5, 7}, as specified in Section 4.5.
>
> [RF3] Wu, Lijun, et al. "R-Drop: Regularized Dropout for Neural Networks." Advances in Neural Information Processing Systems 34 (2021): 10890-10905.
>
> **Result**
>
> According to the table below, R-Drop leads to a slight improvement in scores, but the difference is not significant and remains closely comparable to scores from normal training (Normal).
>
> | Dataset  | Zero-shot | Normal | R-Drop | Denoise |
> |----------|-----------|--------|--------|---------|
> | FiQA     | 35.0      | 36.4   | 36.3   | **36.7**    |
> | Covid    | 75.4      | 77.5   | 78.1   | **78.6**    |
> | SciFact  | 68.8      | 70.6   | 70.8   | **71.1**    |
> | Avg.     | 59.7      | 61.5   | 61.7   | **62.1**    |
>
>
> > Is the validation set available? If not, is the performance reported for the last checkpoint?
>
> We reported our results based on the checkpoint with the highest evaluation scores during the training process.
> + For TREC-COVID and SciFact datasets, as there are no official dev sets, we conducted testing on the test set every 1000 training steps, re-ranking the BM25 top 100 results using NDCG@10 as the evaluation metric.
> + For the FiQA dataset, since an official dev set is available, we directly used the dev set's queries. Similarly, during training, we evaluated the model every 1000 training steps by re-ranking the BM25 top 100 results using NDCG@10.
>
> **To Reviewer:**
> Once again, we appreciate your constructive feedback and suggestions, which helps us to enhance the comprehensiveness of our research.

---

### Official Review · Reviewer_zViP · 2023-08-19

**Soundness:** 4

**Excitement:**

4: Strong: This paper deepens the understanding of some phenomenon or lowers the barriers to an existing research direction.

**Paper Topic And Main Contributions:**

This paper proposes a new method for unsupervised domain adaptation for both the dense retrieval model and rerank model in a two-stage retrieval system. It first constructs a pseudo training set and then denoise-finetuning the rerank model. It finally distills the knowledge from the rerank model to the dense retrieval model. The method achieves good performance improvement in several benchmarks. The ablation studies also demostrate the effectiveness of differetn components.

**Questions For The Authors:**

- How does the overall training cost compared to the baseline methods? The proposed method utilizes several separate steps to adapt the two components in a retrieval system. The training procedure might be more complicated and requires lots of tuning efforts?

Update: The authors have provided the training cost in GPU hours. The method does require longer training time, but as the authors claimed, it is a reasonable investment to improve the overall performance.

**Reasons To Accept:**

- The paper is well written and easy to understand.
- The proposed method adapts both the rerank model and the retrieval model without labeled target data. It also adopts denoise-finetuning to mitigate the impact of noisy labels. This technique is overall novel.
- The proposed method shows good performance and the ablation studies demonstrate the effectiveness.

**Reasons To Reject:**

The method has three steps which can be complicated and can require more tuning efforts.

**Reproducibility:**

4: Could mostly reproduce the results, but there may be some variation because of sample variance or minor variations in their interpretation of the protocol or method.

**Reviewer Confidence:**

2: Willing to defend my evaluation, but it is fairly likely that I missed some details, didn't understand some central points, or can't be sure about the novelty of the work.

**Typos Grammar Style And Presentation Improvements:**

- line 195: a space is missing before "("
- line 322: "label" should be subscript

---

> ### Author Rebuttal · Authors · 2023-08-28
>
> Thank you for your insightful review of our paper.
>
>
> > How does the overall training cost compared to the baseline methods? The proposed method utilizes several separate steps to adapt the two components in a retrieval system. The training procedure might be more complicated and requires lots of tuning efforts?
>
> You've raised a important question regarding the training costs and complexity compared to baseline methods.
>
> Our primary comparison target is the GPL[RF1]  method. In comparison to the baseline method GPL, the steps are as follows:
> 1. Generating queries for passages (~5hr)
> 2. In the distillation phase, labeling using the cross-encoder (~4hr for labeling)
> 3. Training the bi-encoder (~10hr)
>
> Total: 19hr
>
> [RF1] Wang, Kexin, et al. "GPL: Generative Pseudo Labeling for Unsupervised Domain Adaptation of Dense Retrieval." Proceedings of the 2022 Conference of the North American Chapter of the Association for Computational Linguistics: Human Language Technologies. 2022.
>
> In our approach, building upon GPL, we further finetune the cross-encoder. This introduces the following steps:
> 1. Apply domain adaptation on cross-encoder.
> 2. Labeling with both the original cross-encoder CE(w/Ad) and the adapted CE(wo/Ad) cross-encoder, which adds extra labeling time but doesn't impact bi-encoder training.
>
> Hence, the training process involves these stages:
> 1. Generating queries for passages (~5hr)
> 2. Preparing training data for cross-encoder and conducting denoise fine-tuning (~2hr)
> 3. Labeling using both the cross-encoder and the adapted cross-encoder (~4hr + 4hr)
> 4. Training the bi-encoder (~10hr)
>
> Total: 25hr
>
> All our experiments were performed on a single RTX 3090 GPU. Utilizing multiple GPUs would accelerate computation speed. It's important to note that inference times remain consistent. Even during denoise fine-tuning, we exclusively employ the model that demonstrates superior performance on test scores for inference and distillation purposes.
>
> We acknowledge the increased training time as a consideration. However, the additional steps we introduce, such as cross-encoder training and using adapted cross-encoders for labeling, contribute to the effectiveness of our approach in achieving domain adaptation. We believe that the enhanced performance justifies the additional training investment.
>
> > line 195: a space is missing before "("
> line 322: "label" should be subscript
>
> Thank you for pointing out these issues! We will add the missing space and correct the subscript for "label" as suggested.
>
> Once again, we appreciate your constructive feedback and suggestions.

---

### Meta-Review · Area_Chair_fjBQ · 2023-09-27

**Recommendation:** 2

**Metareview:**

This paper presents a novel approach to unsupervised domain adaptation for both the dense retrieval model and rerank model in a two-stage retrieval system. Through meticulous experimentation, the authors validate the efficacy of their proposed method in domain adaptation. The reviewers' opinions on this work are not unanimously positive. While the author has addressed certain concerns raised by the reviewers, there are still some lingering apprehensions. zViP expressed concerns about the training time consumption of the method. The authors responded by providing a detailed comparison with previous methods and indicating the time consumption involved. Although the method does require a significant amount of time, zViP acknowledges that these costs are a reasonable investment. X5Zx raised concerns regarding method details, experimental details, and experimental repeatability. The authors provided a thorough response addressing these concerns. kWur recognizes the novelty of the method and provides suggestions for further revision of the paper. XVNc expressed concerns about the weaknesses in the comparison methods, the complexity of calculations leading to practical feasibility issues, and difficulties in reproducibility. The authors provided corresponding responses, but these concerns still remain.

---

### Decision · Program_Chairs · 2023-10-07

**Decision:**

Accept-Findings

**Comment:**

This paper presents a novel approach to unsupervised domain adaptation for both the dense retrieval model and rerank model in a two-stage retrieval system. Through meticulous experimentation, the authors validate the efficacy of their proposed method in domain adaptation. The reviewers' opinions on this work are not unanimously positive. While the author has addressed certain concerns raised by the reviewers, there are still some lingering apprehensions. zViP expressed concerns about the training time consumption of the method. The authors responded by providing a detailed comparison with previous methods and indicating the time consumption involved. Although the method does require a significant amount of time, zViP acknowledges that these costs are a reasonable investment. X5Zx raised concerns regarding method details, experimental details, and experimental repeatability. The authors provided a thorough response addressing these concerns. kWur recognizes the novelty of the method and provides suggestions for further revision of the paper. XVNc expressed concerns about the weaknesses in the comparison methods, the complexity of calculations leading to practical feasibility issues, and difficulties in reproducibility. The authors provided corresponding responses, but these concerns still remain.